# Label-free determination of diffusion coefficients at the nanoscale through modelling of the Surface Plasmon Resonance signal

**Gabriele Antonio Zingale**[1], **Irene Pandino**[1], **Damiano Calcagno**[1], **Maria Luisa Perina**[2], **Nunzio Tuccitto**[2], **Giuseppe Grasso**[2]*

1 IRCCS-Fondazione Bietti, Rome, Italy, 2 Department of Chemical Sciences, University of Catania, Catania, Italy

☯ These authors contributed equally to this work.

* grassog@unict.it

**Data Availability Statement:** All relevant data are within the paper.

## Abstract

Surface plasmon resonance (SPR) is normally used to measure the kinetic parameters of biomolecular interactions between a molecule immobilized on a gold surface and another one flowing in a microfluidic channel above the surface. During the SPR measurements, convection-diffusion phenomena occur inside the microfluidic channels, but they are generally minimized by appropriate experimental setup in order to obtain diffusion free kinetic parameters of the molecular interactions. In this work, for the first time, a commercial SPR apparatus has been used to obtain non canonical scientific parameters. Indeed, a specifically designed SPR experimental setup is described for carrying out measurements of the diffusion coefficient ($D$) of molecules in solutions. The high precision and reproducibility of the approach, as well as the wide applicability of the newly proposed SPR based method for the measurement of $D$ of many different molecules and biomolecules, are here demonstrated and illustrated in detail.

## Introduction

Surface Plasmon Resonance (SPR) is an analytical technique normally used to study biomolecular interactions occurring between a biomolecule immobilized on a gold surface (ligand) and another one (analyte) flowing inside a flow cell [1, 2]. Although many different experimental conditions can be tailored to obtain specific interaction parameters [3], two interacting entities must be considered in a typical SPR experiment. Indeed, various improved SPR approaches have been developed in order to increase sensitivity and reproducibility in the detection of specific analytes. For example, very recently, WaveFlex biosensors have been applied for tyramine [4] and histamine [5] detection, demonstrating advantages in comparison to a standard SPR system. However, even in these improved SPR approaches, the investigated analyte must interact with a bound ligand in order to be detected.

**Funding:** This research was supported by MIUR, PRIN: P2022AW2H9 "Molecular details on the early phase of amyloid beta peptides aggregation: a multilevel approach based on carbon dots fluorescence and diffusion coefficients measurements to unveil the pathogenic molecular mechanisms at the base of Alzheimer's disease" and "Progetto Pharma-HUB - HUB per il riposizionamento di farmaci nelle malattie rare del sistema nervoso in età pediatrica" (T4-AN-04). The authors further acknowledge LazioInnova for its financial support (grant: A0375-2020-36591). The funders had no role in study design, data collection and analysis, decision to publish, or preparation of the manuscript. • MIUR, PRIN: P2022AW2H9 • Pharma-HUB - HUB per il riposizionamento di farmaci nelle malattie rare del sistema nervoso in età pediatrica" (T4-AN-04) • LazioInnova grant: A0375-2020-36591 The funders had no role in study design, data collection and analysis, decision to publish, or preparation of the manuscript."

**Competing interests:** The authors have declared that no competing interests exist.

**Abbreviations:** ATP, adenosine triphosphate; BSA, bovine serum albumin; D, diffusion coefficient; $D_2O$, heavy water; DMSO, Dimethyl sulfoxide; DTPS, Di(N-succinimidyl) 3,3′-dithiodipropionate; IEP, isoelectric point; NHS, N-hydroxysuccinimide; PBS, Phosphate Buffered Saline; SAM, self-assembled monolayer; SPR, Surface Plasmon Resonance; TRIS, tris(hydroxymethyl) aminomethane.

In a new perspective, we have recently demonstrated that SPR can be also applied to investigate some molecular properties of a single biomolecule, without any interaction occurring on the SPR gold surface [6, 7]. Particularly, the diffusion coefficient (D) is a parameter widely investigated by the scientific community because of the important information which are strictly correlated to it [8]. Indeed, D depends not only on the size of the molecular entity under investigation, but also on its shape, as well as on the environment where the diffusion is taking place. In the case of biomolecules, it is even possible to correlate the D values with conformational and/or oligomerization information [7]. The most common way to correlate D with the hydrodynamic radius (r) of the molecules is to consider the widely accepted and well-known Stokes–Einstein–Sutherland equation, for diffusion of spherical particles through a liquid with low Reynolds number:

$$D = \frac{k_B T}{6\pi\eta r} \tag{1}$$

Where $k_B$ is the Boltzmann constant, $T$ is the temperature, and $\eta$ is the dynamic viscosity of the fluid. However, in many cases, the molecules cannot be assimilated to a sphere and more complex equations must be considered. In addition, if the solution is not infinitely dilute and interparticle forces between different solute molecules have to be considered, further adjustments of (1) must be taken into account. However, what could seem a limitation, in turn translates into a great opportunity. In other words, at least in principle, it should be possible to obtain molecular shapes and intermolecular interactions from D measurements [9]. In order to do so, well-designed experiments to accurately measure D values must be performed and some experimental approaches are already present in the current literature. Indeed, nuclear magnetic resonance (NMR) based techniques, computational calculations and methods using light absorption or fluorescence have all been applied to obtain D values in a reliable and accurate way, but many downsides are commonly associated with such experimental approaches. For example, absorption or fluorescence-based techniques require the presence of an absorbent or fluorescent group in the selected wavelength region, so very often molecules must be artificially labelled with it, potentially altering the molecular structure and properties, and therefore the measured D. Analogously, Diffusion-Ordered NMR spectroscopy (DOSY–NMR) technique is limited by the use of aprotic or deuterated solvents, being also not applicable in many cases (according to the Stokes-Einstein equation, D depends on the fluid viscosity, which is affected by the presence of deuterium).

In this perspective, a decade ago Loureiro et al. [10] attempted to use SPR to determine the D values of ethanol and BSA. However, the method did not have any follow-ups, mainly due to the lack of reproducibility and to the difficulties to experimentally perform the measurements. Indeed, the latter required specific experimental conditions (very high flow rates, introduction of air bubbles, large tubes diameters and long experimental times) as well as a customized SPR device. As an example, even the estimation of the ethanol D value was flawed and difficult to be obtained ($0.65 \pm 0.14 \times 10^{-9}$ m²/s) for the concentration therein used (0.85 M).

After more than ten years since the Loureiro et al. paper, we have finally managed to set up a newly developed SPR based method which proved to be highly reliably to obtain very precise and reproducible D values for many different kinds of molecules. The herein proposed method offers several advantages. Firstly, it is label-free, eliminating the need for artificial modifications that could influence D. Moreover, compared to methods that utilize a conventional Taylor dispersion analysis, like the absorption or fluorescence ones, the detection is confined to a single SPR flow-cell instead of two detection windows to monitor the spread of the analyte over time, simplifying the experimental setup. Secondly, the measurements are fast, require

low sample consumption, and can be analysed using a Python script, making the process highly efficient and user-friendly compared to the other methods previously cited. Besides the proof of concept, we have also applied such novel method to give an insight on some important issues such as the monitoring of insulin oligomerization states upon changes of environmental factors [7]. In this paper, our newly designed experimental and calculating model, herewith named *D*-SPR, are described in details in order to be potentially adopted by any commercial SPR users. The results obtained for several molecules with different masses and chemical properties are reported and discussed, demonstrating the applicability of this novel SPR based methodology on a very wide range of molecules.

## Materials and methods

The protocol described in this peer-reviewed article is included for printing as (S1 File) with this article. The protocol described in this peer-reviewed article is published on protocols.io (DOI: dx.doi.org/10.17504/protocols.io.x54v927rml3e/v1) and is included for printing purposes as S1 File.

### Expected results

**1. Data analysis.** For an optimal mathematical analysis, the starting dataset must contain a fairly high amount of points per unit of time. Put simply, it is recommended to make sure having at least one acquisition per second. In the method herein described, the instrument was set to record an SPR response point roughly every 1 s.

The *D* measurement in the protocol presented here is based on Taylor's law of diffusion. This method is widely used to estimate the *D* value of molecules or particles travelling in confined spaces, such as capillary tubes. The principle was first introduced by Sir Geoffrey Taylor in 1953 [11] and later extended to describe the dispersion of solutes in a laminar flow through cylindrical tubes. Taylor-Aris dispersion, as it is often called, combines the effects of molecular diffusion to analyze the longitudinal diffusion of solute plugs in a tube. The basis of Taylor's law is the observation that solutes moving in a fluid medium undergo both axial (along the length of the tube) and radial (across the cross-section of the tube) diffusion. In laminar flow, molecules near the center of the tube move faster than those near the walls, due to the parabolic velocity profile typical of fluid flow in capillaries. This difference in flow velocity causes solutes to disperse longitudinally. However, radial diffusion (which is perpendicular to the direction of flow) acts to mix the faster-moving molecules in the central regions with slower-moving molecules near the tube walls, moderating longitudinal dispersion over time. Taylor's analysis showed that when molecules undergo random movement (radial diffusion), this leads to a concentration profile along the length of the tube, which eventually widens due to axial dispersion. Taylor derived an effective longitudinal *D* incorporating both molecular diffusion and convective flow effects. In our experimental protocol, Taylor's diffusion analysis is used to estimate the *D* value of solute molecules by analysing their diffusion behaviour in the capillary tube. In this case, a microfluidic system is used in which the fluid is pushed towards the microfluidic cell of the SPR detector. Taylor's law provides a framework for understanding the coupling between convective drift and molecular diffusion in a capillary tube. This interaction is fundamental to analysing solute diffusion, allowing us to calculate a *D* value that takes into account both fluid velocity and the molecular properties of the solute. The assumption that radial diffusion dominates over axial diffusion allows us to simplify the analysis and treat the system with one-dimensional approximations. This greatly reduces the complexity of the model while still providing accurate results for *D* [12]. The length of the capillary tube determines the time and distance over which molecules can diffuse axially. If the tube is relatively

short, the molecules will have limited axial diffusion before reaching the end of the tube. On the other hand, if the tube is long, the molecules will have more opportunity to diffuse axially. By neglecting axial diffusion, the movement of particles in the capillary tube due to the radial diffusion can then be described by Fick's second law of diffusion, which relates the rate of diffusion to the concentration gradient. Taylor diffusion analysis is employed to estimate the $D$ value by analyzing the spreading behaviour of solute molecules in the capillary tube experimentally, through SPR measurements and fitting the data to a model. The concentration profile after a given time $t$ is commonly referred to the time-dependent Gaussian distribution as

$$C \propto C_0 e^{\left(\frac{-tr^2}{2\sigma^2}\right)}$$

where $tr$ is the time that it takes for the plug to reach the detector placed at the end of the tube. During the traveling, radial diffusion occurs into the tube, thus $\sigma$ is the variance of the Gaussian shaped concentration profile at the detector. The derivation of the equation is based on the assumption that the initial concentration $C_0$ is located in an infinitesimally small region at the point $r = 0$. Since this is quite difficult to achieve during an SPR experiment, we used a different approach. We release a plug of molecules large enough to obtain a cassette like concentration profile (as depicted in S3 Fig in S1 File). Using air bubbles trapping, we are able to obtain, at time $t = 0$ and $r = 0$, a sharp interface between the plug containing the diffusing molecules and the carrier liquid. This interface, initially sharp, undergoes distortion due to the field of laminar forces and radial dispersion, resulting in a concentration profile detected by the SPR detector, similar to an error function (as depicted in S3 Fig (right) in S1 File). By calculating the numerical first derivative, the desired Gaussian shape is obtained and the Taylor's theory can be applied. In Taylor diffusion analysis, the concentration of molecules is typically measured at different positions along the capillary tube, and the diffusion equation is solved to determine the $D$ value. This can be done by fitting the experimental concentration profile to the theoretical solution of the diffusion equation. The resulting $D$ provides information about the mobility and diffusive behaviour of particles within the capillary tube. It is an important parameter in various fields, such as fluid dynamics, microfluidics, and chromatography, where the movement of particles in confined spaces is of interest.

Taylor diffusion analysis offers a reliable framework for measuring the $D$ values of solute molecules in capillary tubes. However, the method assumes idealized conditions, such as a fully developed flow profile and an adequate capillary length to ensure the correct behaviour of solutes in both radial and axial diffusion. In practice, the finite tube length and the high flow velocity often lead to deviations from ideal assumptions, which introduce asymmetries in the detected concentration profile. Recognizing and correcting these deviations is crucial to ensure the accuracy of the derived $D$ values. Under the experimental conditions under which the protocol is performed, one of the main causes of skewness or asymmetry in the diffusion curve is related to insufficient tube length and high flow velocity. When the capillary tube is too short in relation to the velocity of the carrier fluid, the solute cap does not have sufficient time to diffuse completely radially before reaching the detector. This incomplete radial mixing results in an elongated and asymmetric concentration profile, instead of the ideal Gaussian shape expected under full Tayloristic diffusion conditions. The short tube length limits the time available for solutes to equilibrate radially, causing a distortion of the concentration profile detected by the SPR sensor. The relatively high flow velocity exacerbates the problem by rapidly conveying solutes along the axial direction, further reducing the effective radial diffusion time. Both factors result in a non-Gaussian profile at the detector, with an obvious tailing effect on the signal. Given these non-idealities, it is essential to apply appropriate correction factors to ensure that the derived scattering coefficients are accurate. The corrections help to account

for signal skewness due to imperfect mixing of solutes within the capillary. When the concentration profile at the detector is skewed due to short tube length and high flow velocity, a correction is applied to the measured variance of the derivative before the concentration profile. This adjustment compensates for the fact that the solute molecules have not undergone complete radial diffusion, which would normally result in a symmetrical, Gaussian profile. The correction factor adjusts the variance according to the ratio of tube length to flow velocity, ensuring that the *D* value is correctly determined.

To facilitate the data processing and to obtain the calculation of *D*, the authors include here a Python based script. The script in Python is supplied with additional material. The SPR data is exported from BioNavis proprietary software and imported into an Excel sheet. The code also enables data from different replicates to be processed sequentially. The data must be imported into Excel (.xls format) by including the SPR time and response in contiguous columns. An example dataset is provided as additional material. The code includes the Pandas library for importing the dataset from the Excel file, the Numpy library for linear algebra calculations, the Matplotlib library for plotting the data, the Scipy library for curve fitting, the OS library for folder and file management and the Tkinter library for the graphical interface.

First the GUI ask for the Excel file to be processed (Fig 1):

The SPR signal is imported and converted to Numpy array and *nan* values are replaced by 0 using the following code:

*SPR_data = pd.read_excel(fname)*

*dataset = SPR_data.values*

*dataset = np.nan_to_num(dataset, nan = 0)*

A double rows Matplotlib plot appears (Fig 2). The upper plot reports the SPR signal and the lower one the derivative of the signal. A reduced portion of file can be selected and displayed by clicking at the desired slicing time.

The derivative of the signal is fitted with the Exponential-Centred Skew-Normal Distribution and reported as dashed purple line.

$$PDF\_EG(x) = \frac{A}{\tau} e^{\frac{\sigma^2}{2\tau}\frac{x-\mu}{\tau}} \Phi\left(\frac{x-\mu}{\sigma} - \frac{\sigma}{\tau}\right)$$

*def PDF_EG(x, \*p):*

*A, mu, sigma, tau = p*

*return (A/tau)\*(np.exp(0.5\*(sigma\*\*2/tau)-((x-mu)/tau)))\*(1+special.erf(((x-mu)/sigma)-(sigma/tau))*

*coeff, var_matrix = curve_fit(PDF_EG, time, derivative, p0 = p0, absolute_sigma = True)*

where *x* represents time, *tau* is the parameter related to the exponential decay, and *A*, *mu*, and *sigma*, are amplitude, position, and width of the Gaussian, respectively.

The minimized parameters are collected:

*A_f = coeff[0]*

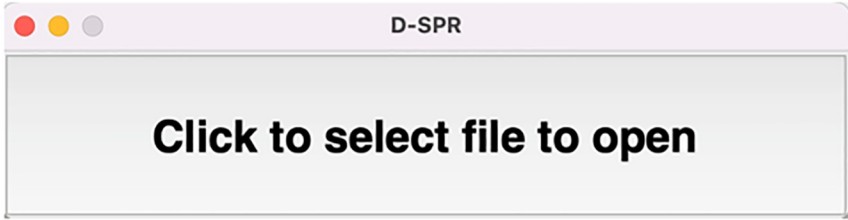

**Fig 1. Screenshot of the GUI generated button for file selection.**

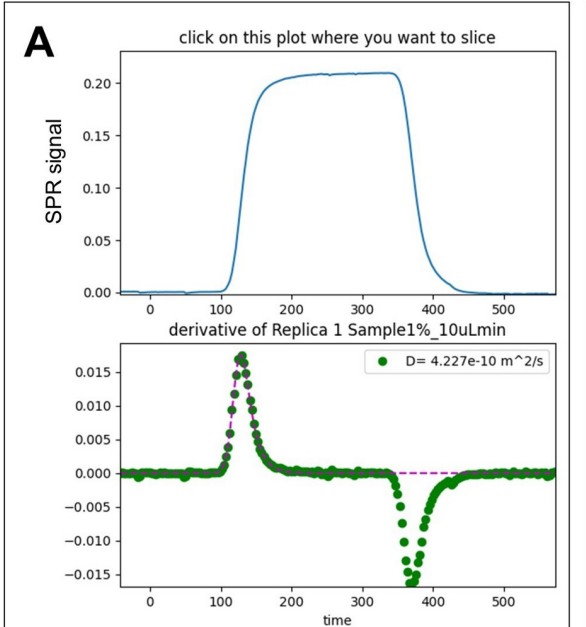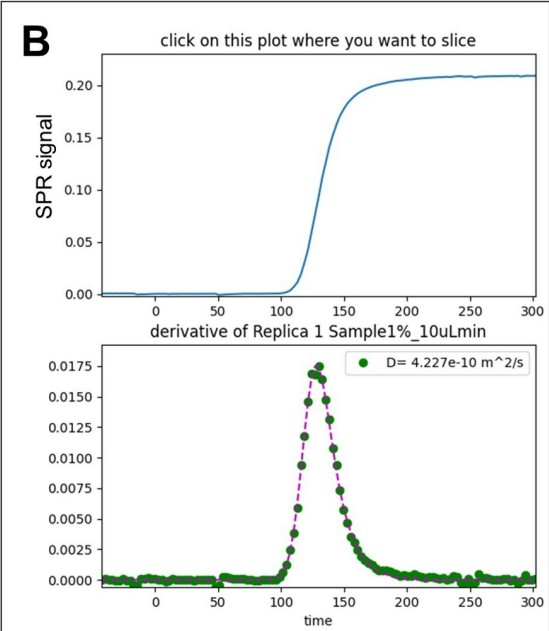

**Fig 2. Example of the Matplotlib plots generated by the script.** (**A**) SPR signal as a function of time (top) and its first derivative plot (bottom). (**B**) Zoom in of the selected SPR signal (top) and its first derivative plot slice (bottom).

*mu_f = coeff[1]*
*sigma_f = coeff[2]*
*tau_f = coeff[3]*
The *D* is calculated:
*m = mu_f + tau_f*
*sq = sigma_f\*\*2 + tau_f\*\*2*
*sk = 2\*(tau_f/sigma_f)\*\*3/((1+(tau_f/sigma_f)\*\*2)\*\*(3/2))*
*alfa = 0.93*
*D = alfa\*(0.127\*\*2\*m/(24\*sq)\*1E-6\*(1+sk))*

Where *alpha* is a calibration parameter that is obtained experimentally. Although the algorithm implemented in the code reported here is slightly different with respect to the one reported in [6] for efficiency purpose, results given are equivalent. The calculated *D* is displayed on the derivative curve label in the plot. The code also provides a text file including *D* and the fitting parameters (Fig 3). After closing the plot window, the code continues calculating D if the imported Excel file includes data from further replicas.

**2. Method validation.** To validate the method herein described, the latter was applied for the determination of *D* of the following small molecules: heavy water ($D_2O$), ethanol, glycine,

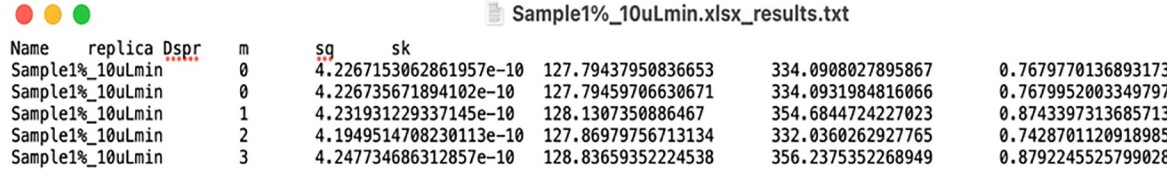

**Fig 3.**

adenosine triphosphate (ATP) and carnosine. The employed molecules do not show any interaction with the gold surface, therefore the signal shown by the instrument is only given by the transition of the sample into the flow cell. On the contrary, $D$ measurements of larger molecules, particularly peptides and proteins, require functionalized gold surfaces to minimize non-specific interactions. Proteins can adhere strongly to gold due to electrostatic, hydrophobic, and non-specific forces. Functionalization strategies herein used involve modifying the gold surface with molecules that have a similar charge distribution to the protein of interest (see section 4 of the S1 File). For example, BSA, a negatively charged protein on most of its surface, is often used to functionalize the gold sensor-chip to minimize interactions of negatively charged proteins. Conversely, poly-L-lysine, positively charged on its surface, is suitable for positively charged proteins. For all of the above listed molecules, milli-Q water was used as running buffer.

The $D$ values of $D_2O$, ethanol, glycine, ATP and carnosine were determined injecting, respectively, a pure $D_2O$, a 1% solution (0.22 M) ethanol in milli-Q water, a 1% glycine solution (0.13 M), pH 6.19, in milli-Q water, a 19.7 mM, pH 5.2, ATP solution in milli-Q water and a 30 mM, pH 7.28, carnosine solution in milli-Q water into the instrument following the steps described in the Methods details section n. 5 in the S1 File. The sigmoidal SPR curves and their 1st derivative are shown in Figs 4–8, respectively.

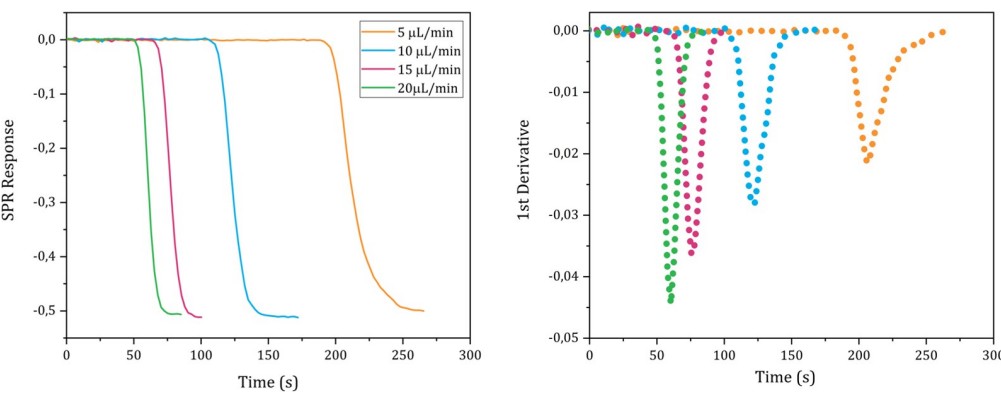

**Fig 4.**

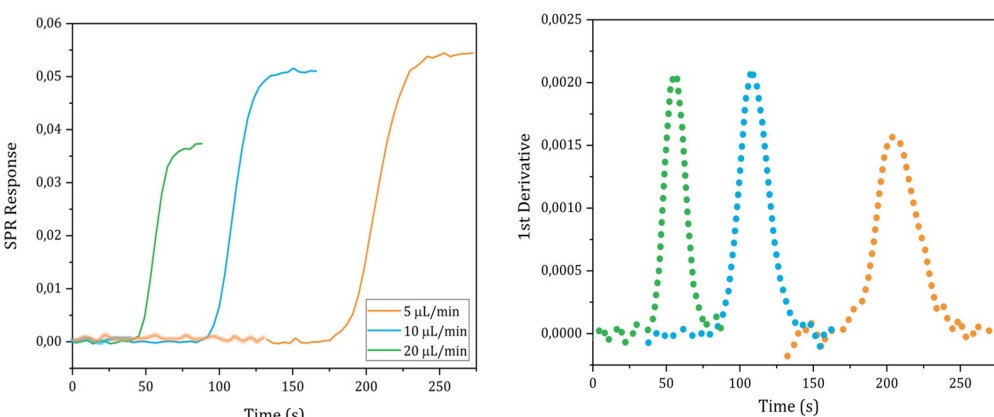

**Fig 5. Plots of the SPR signal (left) and its first derivative (right) of ethanol injections performed at different flow rates.**

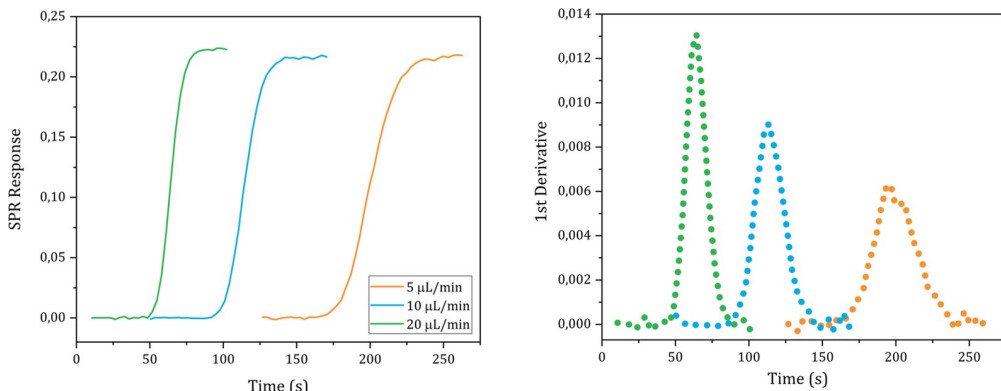

**Fig 6. Plots of the SPR signal (left) and its first derivative (right) of glycine injections performed at different flow rates.**

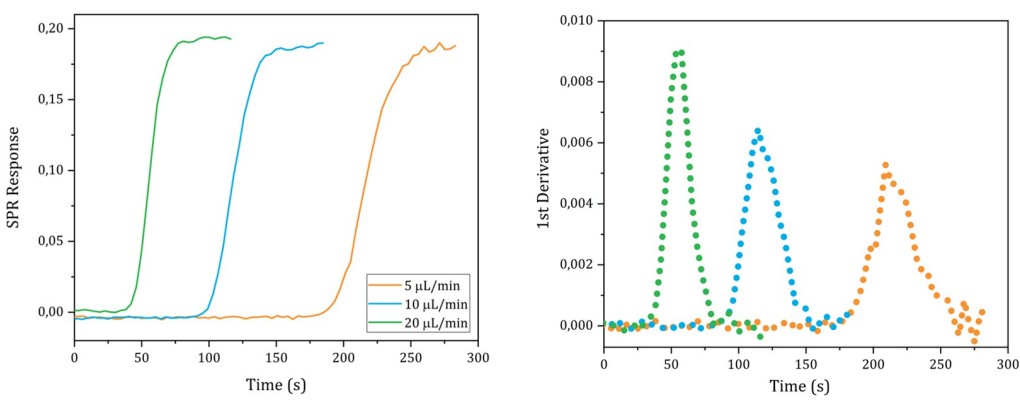

**Fig 7.**

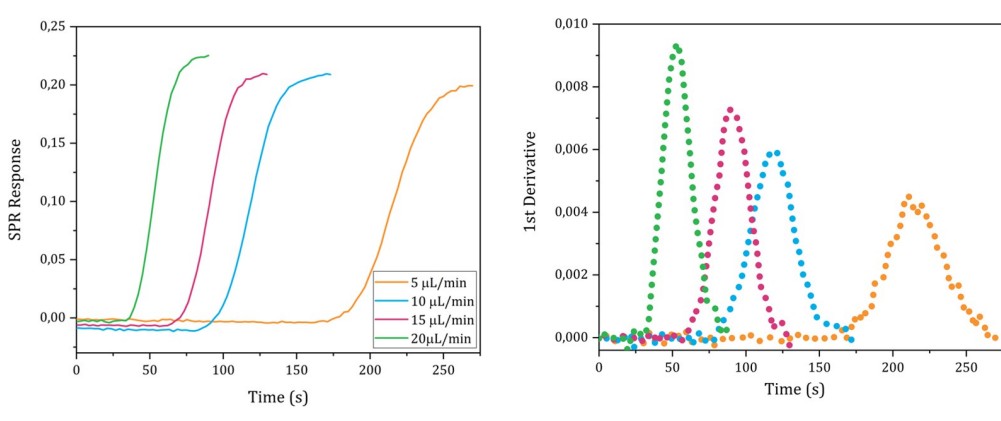

**Fig 8.**

The $D$ values of $D_2O$, ethanol, glycine, ATP and carnosine were calculated applying the mathematical analysis shown in section n. 1 to the experimental data and the values obtained from each technical replicate are shown in Tables 1–5, respectively.

**3. $D$-SPR in protein research.** We addressed limitations associated with analyzing larger biomolecules, such as peptides and proteins, by extending the injection tubing from 35 cm to 70 cm using a union connector. This modification minimized the skewness, resulting in a significantly improved first derivative sensorgram with an ideal Taylor dispersion profile. Consequently, this approach enabled the accurate determination of $D$ for complex biomolecules, including insulin species [7] and lysozyme. Our method offers the unique ability to distinguish between different insulin oligomeric states or isoforms. By analyzing 100 µM insulin solutions under various conditions (pH 3.3, 6.6, and 7.4), with or without zinc and copper ions, we investigated the presence of specific oligomeric forms. The results demonstrate that changes in the equilibrium between various insulin oligomeric states can be readily detected by monitoring variations in the SPR signal. Notably, the first derivative Gaussian distributions exhibits increasing positive asymmetry as insulin progresses towards more complex oligomeric states (Fig 9), which coincides with a decrease in $D$ values calculated by our mathematical model (Table 6). To further validate the effectiveness and reliability of our $D$-SPR system for analyzing macromolecules, we conducted a comprehensive diffusion analysis of lysozyme in an

**Table 1. Data obtained from the analysis of the SPR injections of $D_2O$.**

| Flow rate (µL/min) | $t_r$ (s) | $\sigma^2$ | sk | $D_{SPR}$ (m²/s) | $\bar{D}_{SPR}$ (m²/s) | error (m²/s) |
|---|---|---|---|---|---|---|
| 5 | 192.25 | 198.35 | 0.41 | $1.19 \cdot 10^{-9}$ | $2.02 \cdot 10^{-9}$ | $2.19 \cdot 10^{-10}$ |
| | 185.36 | 195.26 | 0.41 | $1.17 \cdot 10^{-9}$ | | |
| | 193.18 | 215.84 | 0.42 | $1.11 \cdot 10^{-9}$ | | |
| | 186.36 | 213.68 | 0.41 | $1.08 \cdot 10^{-9}$ | | |
| 10 | 121.38 | 62.11 | 0.03 | $1.75 \cdot 10^{-9}$ | | |
| | 119.05 | 55.53 | 0.03 | $1.92 \cdot 10^{-9}$ | | |
| | 122.57 | 65.20 | 0.01 | $1.66 \cdot 10^{-9}$ | | |
| | 123.62 | 68.27 | 0.20 | $1.91 \cdot 10^{-9}$ | | |
| | 121.15 | 60.25 | 0.20 | $2.10 \cdot 10^{-9}$ | | |
| | 124.12 | 65.97 | 0.19 | $1.96 \cdot 10^{-9}$ | | |
| | 121.82 | 63.25 | 0.22 | $2.05 \cdot 10^{-9}$ | | |
| 15 | 75.75 | 32.38 | 0.07 | $2.18 \cdot 10^{-9}$ | | |
| | 79.26 | 34.77 | 0.07 | $2.13 \cdot 10^{-9}$ | | |
| | 76.44 | 32.90 | 0.02 | $2.07 \cdot 10^{-9}$ | | |
| | 80.05 | 35.08 | 0.04 | $2.07 \cdot 10^{-9}$ | | |
| | 77.41 | 35.82 | 0.13 | $2.13 \cdot 10^{-9}$ | | |
| | 81.06 | 37.27 | 0.13 | $2.15 \cdot 10^{-9}$ | | |
| | 77.90 | 35.91 | 0.13 | $2.15 \cdot 10^{-9}$ | | |
| | 81.57 | 38.77 | 0.15 | $2.11 \cdot 10^{-9}$ | | |
| 20 | 59.53 | 22.45 | 0.08 | $2.49 \cdot 10^{-9}$ | | |
| | 62.05 | 22.96 | 0.05 | $2.49 \cdot 10^{-9}$ | | |
| | 60.21 | 22.96 | 0.06 | $2.42 \cdot 10^{-9}$ | | |
| | 62.77 | 24.20 | 0.05 | $2.38 \cdot 10^{-9}$ | | |
| | 60.87 | 24.36 | 0.13 | $2.46 \cdot 10^{-9}$ | | |
| | 63.36 | 24.97 | 0.11 | $2.45 \cdot 10^{-9}$ | | |
| | 61.28 | 24.17 | 0.12 | $2.47 \cdot 10^{-9}$ | | |
| | 63.82 | 25.06 | 0.11 | $2.46 \cdot 10^{-9}$ | | |

**Table 2. Data obtained from the analysis of the SPR injections of ethanol.**

| Flow rate (µL/min) | $t_r$ (s) | $\sigma^2$ | sk | $D_{SPR}$ (m²/s) | $\bar{D}_{SPR}$ (m²/s) | error (m²/s) |
|---|---|---|---|---|---|---|
| 5 | 208.39 | 214.21 | 0.14 | $9.73 \cdot 10^{-10}$ | $1.02 \cdot 10^{-9}$ | $2.51 \cdot 10^{-11}$ |
| | 206.65 | 204.79 | 0.08 | $9.54 \cdot 10^{-10}$ | | |
| | 205.80 | 213.35 | 0.13 | $9.53 \cdot 10^{-10}$ | | |
| 10 | 107.99 | 97.32 | 0.06 | $1.03 \cdot 10^{-9}$ | | |
| | 107.99 | 97.32 | 0.06 | $1.03 \cdot 10^{-9}$ | | |
| | 110.66 | 109.82 | 0.15 | $1.01 \cdot 10^{-9}$ | | |
| | 108.55 | 98.75 | 0.09 | $1.05 \cdot 10^{-9}$ | | |
| 20 | 56.45 | 51.16 | 0.16 | $1.12 \cdot 10^{-9}$ | | |
| | 54.39 | 53.07 | 0.13 | $1.01 \cdot 10^{-9}$ | | |
| | 56.50 | 52.41 | 0.12 | $1.06 \cdot 10^{-9}$ | | |

**Table 3. Data obtained from the analysis of the SPR injections of glycine.**

| Flow rate (µL/min) | $t_r$ (s) | $\sigma^2$ | sk | $D$-SPR (m²/s) | $\bar{D}_{SPR}$ (m²/s) | error (m²/s) |
|---|---|---|---|---|---|---|
| 5 | 204.47 | 227.59 | 0.12 | $8.82 \cdot 10^{-10}$ | $9.97 \cdot 10^{-10}$ | $6.07 \cdot 10^{-11}$ |
| | 203.33 | 237.50 | 0.17 | $8.77 \cdot 10^{-10}$ | | |
| | 214.86 | 211.55 | 0.08 | $9.61 \cdot 10^{-10}$ | | |
| | 208.54 | 217.23 | 0.09 | $9.18 \cdot 10^{-10}$ | | |
| | 200.89 | 241.01 | 0.15 | $8.36 \cdot 10^{-10}$ | | |
| | 199.58 | 229.03 | 0.14 | $8.67 \cdot 10^{-10}$ | | |
| 10 | 110.86 | 109.35 | 0.10 | $9.77 \cdot 10^{-10}$ | | |
| | 109.25 | 115.84 | 0.10 | $9.04 \cdot 10^{-10}$ | | |
| | 109.95 | 110.26 | 0.13 | $9.83 \cdot 10^{-10}$ | | |
| | 114.70 | 103.86 | 0.10 | $1.06 \cdot 10^{-9}$ | | |
| | 119.21 | 128.46 | 0.14 | $9.27 \cdot 10^{-10}$ | | |
| 20 | 59.95 | 52.16 | 0.11 | $1.11 \cdot 10^{-9}$ | | |
| | 61.02 | 51.38 | 0.07 | $1.11 \cdot 10^{-9}$ | | |
| | 61.20 | 52.54 | 0.10 | $1.12 \cdot 10^{-9}$ | | |
| | 63.70 | 51.19 | 0.09 | $1.18 \cdot 10^{-9}$ | | |
| | 64.58 | 51.51 | 0.11 | $1.21 \cdot 10^{-9}$ | | |

aqueous solution (10mM PBS-HCl, pH 3.3). Lysozyme is a well-characterized enzyme with a known $D$, making it an ideal candidate for system validation. The results are presented in Fig 10 and Table 7, providing strong empirical support for our method.

## Conclusions

The method herein described has the advantages to be label free and potentially suitable to every molecule of any size. Furthermore, the approach looks promising for biochemical and biomedical applications. For instance, it may be applied to monitor the conformational states of proteins or peptides in solution as the $D$ value is expected to change accordingly. Additionally, within a complex mixture, as a biological fluid actually is, it may be applied to uncover how the physical properties change under health and disease. With respect to this, fluids with a low protein content such as cerebrospinal fluid or aqueous and vitreous humour (eye fluids researched for pathological parameters of neurodegenerative disorders, including glaucoma,

**Table 4. Data obtained from the analysis of the SPR injections of ATP.**

| Flow rate (µL/min) | $t_r$ (s) | $\sigma^2$ | sk | $D_{SPR}$ (m²/s) | $\bar{D}_{SPR}$ (m²/s) | error (m²/s) |
|---|---|---|---|---|---|---|
| 5 | 212.65 | 291.14 | 0.14 | $7.31 \cdot 10^{-10}$ | $7.11 \cdot 10^{-10}$ | $2.74 \cdot 10^{-11}$ |
| | 211.48 | 288.23 | 0.09 | $6.98 \cdot 10^{-10}$ | | |
| | 212.64 | 291.35 | 0.11 | $7.07 \cdot 10^{-10}$ | | |
| | 219.02 | 308.75 | 0.15 | $7.10 \cdot 10^{-10}$ | | |
| 10 | 112.16 | 170.21 | 0.11 | $6.41 \cdot 10^{-10}$ | | |
| | 115.81 | 164.81 | 0.10 | $6.76 \cdot 10^{-10}$ | | |
| | 122.78 | 167.88 | 0.15 | $7.38 \cdot 10^{-10}$ | | |
| | 113.32 | 179.53 | 0.15 | $6.34 \cdot 10^{-10}$ | | |
| | 116.44 | 158.79 | 0.11 | $7.10 \cdot 10^{-10}$ | | |
| | 123.39 | 160.15 | 0.13 | $7.59 \cdot 10^{-10}$ | | |
| | 115.13 | 175.03 | 0.13 | $6.47 \cdot 10^{-10}$ | | |
| | 118.82 | 176.95 | 0.15 | $6.77 \cdot 10^{-10}$ | | |
| | 125.63 | 175.01 | 0.19 | $7.47 \cdot 10^{-10}$ | | |
| | 116.08 | 174.05 | 0.15 | $6.71 \cdot 10^{-10}$ | | |
| | 119.15 | 161.31 | 0.11 | $7.13 \cdot 10^{-10}$ | | |
| | 126.26 | 164.04 | 0.14 | $7.67 \cdot 10^{-10}$ | | |
| 20 | 57.26 | 88.60 | 0.16 | $6.54 \cdot 10^{-10}$ | | |
| | 55.61 | 85.06 | 0.16 | $6.62 \cdot 10^{-10}$ | | |
| | 66.61 | 80.02 | 0.14 | $8.32 \cdot 10^{-10}$ | | |
| | 57.73 | 82.70 | 0.16 | $7.06 \cdot 10^{-10}$ | | |
| | 56.35 | 86.69 | 0.17 | $6.67 \cdot 10^{-10}$ | | |
| | 67.51 | 84.72 | 0.18 | $8.20 \cdot 10^{-10}$ | | |
| | 58.24 | 87.68 | 0.18 | $6.87 \cdot 10^{-10}$ | | |
| | 56.61 | 82.66 | 0.17 | $7.01 \cdot 10^{-10}$ | | |
| | 68.19 | 92.54 | 0.21 | $7.82 \cdot 10^{-10}$ | | |
| | 58.80 | 89.31 | 0.17 | $6.73 \cdot 10^{-10}$ | | |
| | 57.01 | 82.62 | 0.13 | $6.80 \cdot 10^{-10}$ | | |
| | 68.21 | 82.13 | 0.13 | $8.20 \cdot 10^{-10}$ | | |

Alzheimer's disease, etc.), and for which a pilot investigation has been already run, are worth being assayed for the global $D$ value as a physical parameter of their composition. In accordance, with the perspective on the applications here discussed, at least for the tested molecules, the method has proven to produce highly reproducible and reliable results for the estimation of the $D$ value and can be applied in a wide range of experimental conditions. Problems regarding the inertness of the surface, that is possible adhesion of the analyte to the SPR gold chip might arise, but the usage of an appropriate and specific chemical preparation of the gold surface, like the ones here proposed, should circumvent any possible limitations.

The importance of $D$ values in biomolecular analysis cannot be overstated. $D$ provides valuable insights into the size, shape, and overall conformation of a biomolecule. Additionally, by analyzing the changes in $D$ values upon interaction with another molecule or under varying conditions (like pH or presence of metal ions), we can gain a deeper understanding of the binding affinity and kinetics. This information is critical for drug discovery efforts, where researchers aim to develop molecules that specifically target and interact with desired proteins. For instance, in drug delivery, understanding drug diffusion properties is essential for designing efficient systems. A drug with a high $D$ can more readily penetrate tissues, enhancing its bioavailability. Furthermore, $D$ can provide insights into drug-target interactions too. Indeed,

**Table 5. Data obtained from the analysis of the SPR injections of carnosine.**

| Flow rate (μL/min) | $t_r$ (s) | $\sigma^2$ | sk | $D_{SPR}$ (m²/s) | $\bar{D}_{SPR}$ (m²/s) | error (m²/s) |
|---|---|---|---|---|---|---|
| 10 | 115.40 | 198.26 | 0.01 | $5.14 \cdot 10^{-10}$ | $5.43 \cdot 10^{-10}$ | $2.41 \cdot 10^{-11}$ |
| | 120.85 | 198.21 | 0.01 | $5.39 \cdot 10^{-10}$ | | |
| | 116.76 | 200.89 | 0.01 | $5.14 \cdot 10^{-10}$ | | |
| | 121.96 | 190.99 | 0.01 | $5.65 \cdot 10^{-10}$ | | |
| | 118.48 | 215.53 | 0.01 | $4.87 \cdot 10^{-10}$ | | |
| | 124.34 | 224.09 | 0.07 | $5.17 \cdot 10^{-10}$ | | |
| | 119.60 | 208.09 | 0.01 | $5.08 \cdot 10^{-10}$ | | |
| | 125.23 | 215.00 | 0.06 | $5.38 \cdot 10^{-10}$ | | |
| 15 | 89.66 | 142.80 | 0.04 | $5.68 \cdot 10^{-10}$ | | |
| | 81.77 | 132.44 | 0.11 | $5.97 \cdot 10^{-10}$ | | |
| | 90.81 | 138.65 | 0.07 | $6.14 \cdot 10^{-10}$ | | |
| | 82.56 | 129.03 | 0.12 | $6.25 \cdot 10^{-10}$ | | |
| | 91.04 | 139.30 | 0.02 | $5.85 \cdot 10^{-10}$ | | |
| | 83.27 | 139.08 | 0.14 | $5.94 \cdot 10^{-10}$ | | |
| | 91.78 | 134.13 | 0.02 | $6.08 \cdot 10^{-10}$ | | |
| | 84.11 | 131.37 | 0.11 | $6.23 \cdot 10^{-10}$ | | |
| 20 | 52.84 | 104.96 | 0.07 | $4.72 \cdot 10^{-10}$ | | |
| | 55.40 | 110.72 | 0.09 | $4.76 \cdot 10^{-10}$ | | |
| | 53.77 | 98.30 | 0.10 | $5.25 \cdot 10^{-10}$ | | |
| | 56.36 | 104.97 | 0.11 | $5.18 \cdot 10^{-10}$ | | |
| | 53.80 | 104.81 | 0.11 | $4.96 \cdot 10^{-10}$ | | |
| | 56.40 | 110.14 | 0.11 | $4.96 \cdot 10^{-10}$ | | |
| | 54.71 | 100.54 | 0.09 | $5.18 \cdot 10^{-10}$ | | |
| | 57.26 | 104.91 | 0.09 | $5.21 \cdot 10^{-10}$ | | |

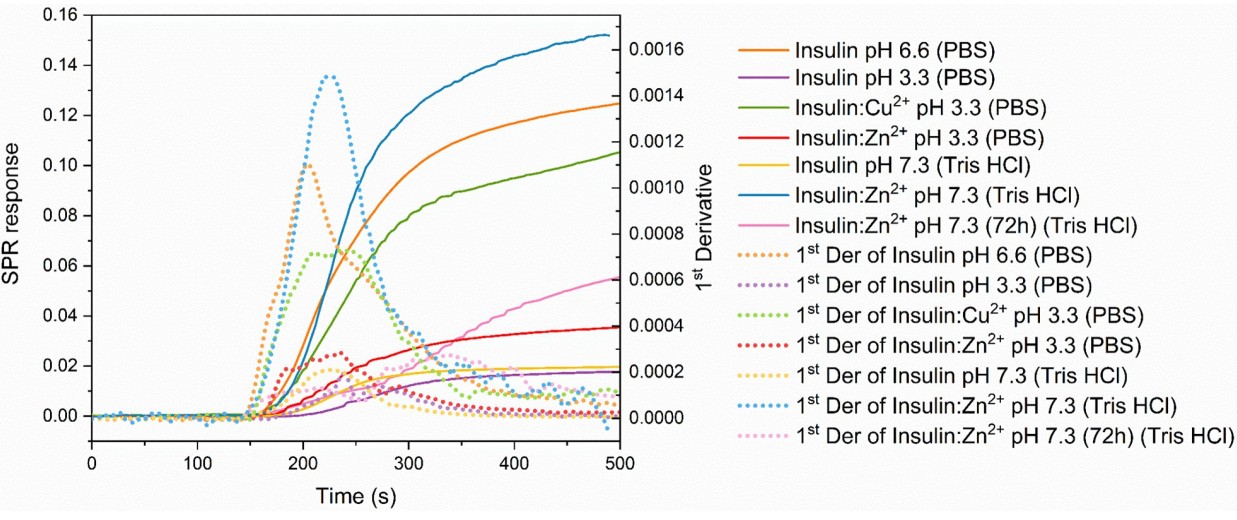

**Fig 9. Experimental *D*-SPR data obtained from the diffusion investigation of insulin under varying experimental conditions are presented: The solid lines in the data correspond to the SPR response resulting from analyte diffusion, while the dotted lines depict the relative 1st derivative.** Insulin solutions were supplemented with zinc or copper ions at a metal-to-peptide ratio 1:1.

**Table 6. Results derived from *D*-SPR analysis at 25˚C:** *D*-SPR values refer to the *D* calculated with the Python script; *error* column reports the standard deviation values and represents the variability within the mean of four experimental replicates.

| Sample | Flow rate (µL/min) | $t_r$ (s) | $\sigma^2$ | sk | $D_{SPR}$ (m²/s) | $\bar{D}_{SPR}$ (m²/s) | error (m²/s) |
|---|---|---|---|---|---|---|---|
| Insulin (pH 3.3 PBS-HCl) | 10 | 263.80 | 2203.05 | 1.61 | $1.55 \cdot 10^{-10}$ | $1.62 \cdot 10^{-10}$ | $1.30 \cdot 10^{-11}$ |
| | | 263.47 | 1816.71 | 1.59 | $1.64 \cdot 10^{-10}$ | | |
| | | 263.67 | 1962.37 | 1.39 | $1.59 \cdot 10^{-10}$ | | |
| | | 264.20 | 1740.39 | 1.51 | $1.73 \cdot 10^{-10}$ | | |
| | | 269.73 | 2218.86 | 1.61 | $1.57 \cdot 10^{-10}$ | | |
| | | 270.65 | 2077.14 | 1.55 | $1.48 \cdot 10^{-10}$ | | |
| | | 265.65 | 1899.00 | 0.03 | $1.88 \cdot 10^{-10}$ | | |
| | | 267.95 | 1971.46 | 1.73 | $1.51 \cdot 10^{-10}$ | | |
| Insulin (pH 6.6 PBS-HCl) | 10 | 226.04 | 2451.00 | 1.87 | $1.11 \cdot 10^{-10}$ | $1.30 \cdot 10^{-10}$ | $1.79 \cdot 10^{-11}$ |
| | | 223.80 | 2255.75 | 1.82 | $1.29 \cdot 10^{-10}$ | | |
| | | 219.99 | 2055.82 | 1.81 | $1.18 \cdot 10^{-10}$ | | |
| | | 229.25 | 2589.71 | 1.87 | $9.68 \cdot 10^{-11}$ | | |
| | | 226.77 | 2362.76 | 1.81 | $1.16 \cdot 10^{-10}$ | | |
| | | 223.09 | 2183.18 | 1.80 | $1.23 \cdot 10^{-10}$ | | |
| | | 220.94 | 1715.05 | 1.86 | $1.41 \cdot 10^{-10}$ | | |
| | | 220.13 | 1678.92 | 1.82 | $1.46 \cdot 10^{-10}$ | | |
| | | 217.06 | 1548.44 | 1.80 | $1.58 \cdot 10^{-10}$ | | |
| | | 222.53 | 1801.75 | 1.87 | $1.35 \cdot 10^{-10}$ | | |
| | | 221.67 | 1732.88 | 1.82 | $1.41 \cdot 10^{-10}$ | | |
| | | 219.13 | 1638.08 | 1.80 | $1.47 \cdot 10^{-10}$ | | |
| Insulin + Zn²⁺ (pH 3.3 PBS-HCl) | 10 | 232.78 | 2276.32 | 1.84 | $1.12 \cdot 10^{-10}$ | $1.17 \cdot 10^{-10}$ | $6.06 \cdot 10^{-12}$ |
| | | 239.08 | 2588.26 | 1.83 | $1.21 \cdot 10^{-10}$ | | |
| | | 241.92 | 2313.71 | 1.74 | $1.16 \cdot 10^{-10}$ | | |
| | | 247.77 | 2528.79 | 1.74 | $1.09 \cdot 10^{-10}$ | | |
| | | 230.94 | 2124.25 | 1.84 | $1.19 \cdot 10^{-10}$ | | |
| | | 237.54 | 2503.16 | 1.89 | $1.28 \cdot 10^{-10}$ | | |
| | | 238.45 | 2225.01 | 1.78 | $1.18 \cdot 10^{-10}$ | | |
| | | 245.15 | 2554.98 | 1.69 | $1.14 \cdot 10^{-10}$ | | |
| Insulin (pH 7.3 TRIS-HCl) | 10 | 228.80 | 1134.62 | 1.35 | $1.67 \cdot 10^{-10}$ | $1.65 \cdot 10^{-10}$ | $1.86 \cdot 10^{-11}$ |
| | | 231.02 | 1164.19 | 1.29 | $1.59 \cdot 10^{-10}$ | | |
| | | 228.53 | 1182.16 | 1.24 | $1.72 \cdot 10^{-10}$ | | |
| | | 229.84 | 1190.30 | 1.24 | $1.88 \cdot 10^{-10}$ | | |
| | | 229.68 | 1211.56 | 1.14 | $1.37 \cdot 10^{-10}$ | | |
| Insulin + Zn²⁺ (pH 7.3 TRIS-HCl) | 10 | 229.35 | 1332.70 | 1.62 | $1.21 \cdot 10^{-10}$ | $1.24 \cdot 10^{-10}$ | $2.34 \cdot 10^{-12}$ |
| | | 229.30 | 1284.66 | 1.60 | $1.26 \cdot 10^{-10}$ | | |
| | | 224.90 | 1164.58 | 1.68 | $1.27 \cdot 10^{-10}$ | | |
| | | 231.99 | 1262.14 | 1.65 | $1.25 \cdot 10^{-10}$ | | |
| | | 235.51 | 1465.63 | 1.62 | $1.23 \cdot 10^{-10}$ | | |
| | | 235.05 | 1388.90 | 1.61 | $1.21 \cdot 10^{-10}$ | | |
| | | 230.11 | 1243.92 | 1.64 | $1.27 \cdot 10^{-10}$ | | |
| | | 237.55 | 1323.49 | 1.62 | $1.25 \cdot 10^{-10}$ | | |

(*Continued*)

**Table 6.** (Continued)

| Sample | Flow rate (μL/min) | $t_r$ (s) | $\sigma^2$ | sk | $D_{SPR}$ (m²/s) | $\bar{D}_{SPR}$ (m²/s) | error (m²/s) |
|---|---|---|---|---|---|---|---|
| Insulin + Zn²⁺ (pH 7.3 TRIS-HCl) 72h | 10 | 349.24 | 12847.19 | 0.71 | $3.06 \cdot 10^{-11}$ | $2.82 \cdot 10^{-11}$ | $1.43 \cdot 10^{-12}$ |
| | | 349.70 | 12632.01 | 0.71 | $3.01 \cdot 10^{-11}$ | | |
| | | 338.02 | 12065.27 | 0.71 | $2.85 \cdot 10^{-11}$ | | |
| | | 352.32 | 13047.10 | 0.71 | $3.04 \cdot 10^{-11}$ | | |
| | | 355.24 | 13054.32 | 0.71 | $2.66 \cdot 10^{-11}$ | | |
| | | 344.80 | 12218.12 | 0.71 | $2.77 \cdot 10^{-11}$ | | |
| | | 357.52 | 15734.88 | 0.71 | $2.75 \cdot 10^{-11}$ | | |
| | | 361.39 | 16242.02 | 0.71 | $2.80 \cdot 10^{-11}$ | | |
| | | 351.22 | 14836.73 | 0.71 | $2.66 \cdot 10^{-11}$ | | |
| | | 357.06 | 14987.48 | 0.71 | $2.68 \cdot 10^{-11}$ | | |
| | | 362.76 | 16229.43 | 0.71 | $2.81 \cdot 10^{-11}$ | | |
| | | 353.55 | 14531.70 | 0.71 | $2.73 \cdot 10^{-11}$ | | |
| Insulin + Cu²⁺ (pH 3.3 PBS-HCl) | 10 | 236.72 | 2352.72 | 0.71 | $1.15 \cdot 10^{-10}$ | $1.18 \cdot 10^{-10}$ | $2.19 \cdot 10^{-12}$ |
| | | 239.65 | 2292.53 | 0.71 | $1.18 \cdot 10^{-10}$ | | |
| | | 239.56 | 2267.75 | 0.71 | $1.19 \cdot 10^{-10}$ | | |
| | | 242.51 | 2216.55 | 0.71 | $1.17 \cdot 10^{-10}$ | | |
| | | 229.45 | 2106.75 | 0.71 | $1.20 \cdot 10^{-10}$ | | |
| | | 231.16 | 1929.11 | 0.71 | $1.18 \cdot 10^{-10}$ | | |
| | | 231.92 | 2160.19 | 0.71 | $1.21 \cdot 10^{-10}$ | | |
| | | 233.80 | 1950.95 | 0.71 | $1.15 \cdot 10^{-10}$ | | |

a drug with a higher $D$ can more effectively reach its intended binding site. Additionally, diffusion properties also influence the drug pharmacokinetics, including absorption, distribution, metabolism, and excretion. By studying diffusion, it is possible to predict the drug behaviour in the body and optimize its formulation.

Additionally, $D$-SPR offers a valuable tool for protein engineering by allowing the assessment of the impact of mutations or modifications on protein structure and function. This approach can enhance the knowledge on existing protein structures, aiding in the prediction of new ones. For example, measuring $D$ of unnatural polypeptides, often created through

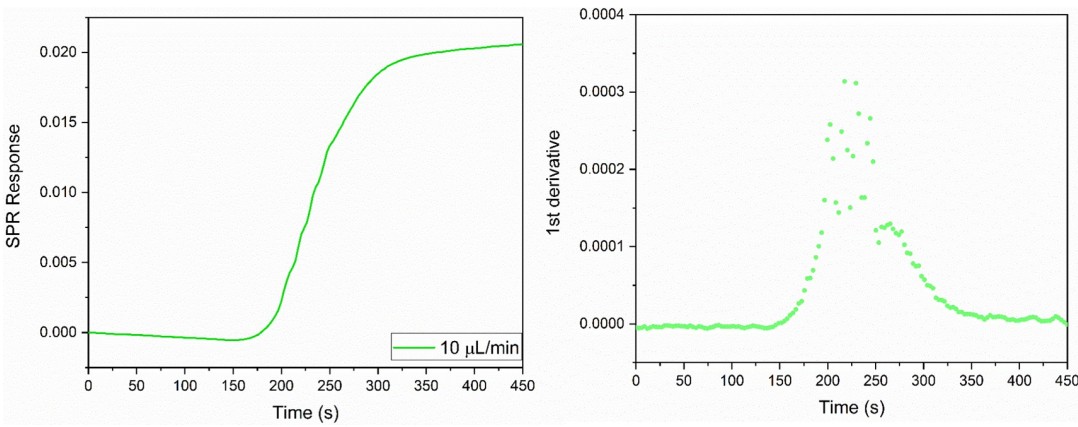

**Fig 10.** $D$-SPR analysis of the Lysozyme solution (349 μM), with the left panel showing the sensorgram curve and the right panel showing the 1st derivative.

**Table 7. Results derived from *D*-SPR analysis of the lysozyme (349μM) at 25˚C:** *D*-SPR values refer to the *D* calculated with the Python script described previously; the *error* column reports the standard deviation values and represents the variability within the mean of four experimental replicates; values are consistent with the ones reported in the existing literature [13].

| Sample | Flow rate (μL/min) | $t_r$ (s) | $\sigma^2$ | sk | $D_{SPR}$ (m²/s) | $\bar{D}_{SPR}$ (m²/s) | error (m²/s) |
|---|---|---|---|---|---|---|---|
| Lysozyme (pH 3.3 PBS-HCl) | 10 | 228.40 | 1304.15 | 0.71 | $1.33 \cdot 10^{-10}$ | $1.45 \cdot 10^{-10}$ | $8.27 \cdot 10^{-12}$ |
| | | 228.80 | 1230.90 | 0.71 | $1.42 \cdot 10^{-10}$ | | |
| | | 229.46 | 1180.24 | 0.71 | $1.48 \cdot 10^{-10}$ | | |
| | | 228.08 | 1131.40 | 0.71 | $1.56 \cdot 10^{-10}$ | | |
| | | 227.74 | 1168.29 | 0.71 | $1.49 \cdot 10^{-10}$ | | |
| | | 229.88 | 1141.55 | 0.71 | $1.53 \cdot 10^{-10}$ | | |
| | | 231.91 | 1355.15 | 0.71 | $1.26 \cdot 10^{-10}$ | | |
| | | 232.28 | 1289.45 | 0.71 | $1.47 \cdot 10^{-10}$ | | |
| | | 232.91 | 1271.26 | 0.71 | $1.50 \cdot 10^{-10}$ | | |
| | | 231.11 | 1190.98 | 0.71 | $1.48 \cdot 10^{-10}$ | | |
| | | 230.66 | 1180.73 | 0.71 | $1.49 \cdot 10^{-10}$ | | |
| | | 233.95 | 1254.06 | 0.71 | $1.42 \cdot 10^{-10}$ | | |

amino acid sequence modifications, pave the way for a wide range of applications. These include engineering proteins by introducing mutations on their structure that alter their *D* for specific purposes. Among the modifications are also included post-translational modifications, such as acetylation, phosphorylation, glycosylation or ubiquitylation at specific residues, that can alter protein diffusion along with their functional role. Furthermore, studying the diffusion of protein complexes can reveal information about their stability and dynamics under various chemical-physical conditions that affect their folding. This connection to protein folding and unfolding allows *D* to be used to investigate conformational changes. For example, comparing the diffusion of a protein before and after unfolding can provide valuable insights into its folding pathway.

In the context of the methodology herein reported, *D*-SPR can be employed to differentiate between various protein conformational and oligomeric states analyzing its diffusion behavior. To assess the sensitivity of our SPR-based approach in detecting minor changes in *D* values, we conducted a series of experiments first on a range of small molecules and then to insulin solution at different pHs, buffers and incubation times. Results, especially the ones in Table 6, show that the *D*-SPR approach is highly sensitive to even subtle changes in *D* at the various experimental conditions. The high sensitivity of this SPR-based approach expands its applicability to a wide range of biological and chemical systems where small changes in diffusion are critical to understanding underlying processes. Additionally, the ability to distinguish subtle changes in insulin oligomerization, as reported in Fig 9 and Table 6, has significant implications for understanding insulin action and potential therapeutic interventions for diabetes. Similarly, the diffusion analysis of lysozyme (Fig 10 and Table 7) validates the effectiveness of *D*-SPR for studying larger macromolecules.

## Supporting information

**S1 File.**
(PDF)

**S1 Graphical abstract.**
(TIF)

## Acknowledgments

G.A. Zingale, I. Pandino and M. L. Perina were supported by the PhD program in Chemical Sciences, University of Catania. The authors acknowledge the Ministry of Health and Fondazione Roma. Figures were created with icons from BioRender.com.

## Author Contributions

**Conceptualization:** Irene Pandino, Nunzio Tuccitto, Giuseppe Grasso.

**Data curation:** Gabriele Antonio Zingale, Irene Pandino, Damiano Calcagno, Nunzio Tuccitto.

**Formal analysis:** Gabriele Antonio Zingale, Irene Pandino, Damiano Calcagno, Nunzio Tuccitto.

**Funding acquisition:** Giuseppe Grasso.

**Investigation:** Gabriele Antonio Zingale, Irene Pandino, Damiano Calcagno, Maria Luisa Perina, Nunzio Tuccitto, Giuseppe Grasso.

**Methodology:** Gabriele Antonio Zingale, Irene Pandino, Damiano Calcagno, Maria Luisa Perina, Nunzio Tuccitto, Giuseppe Grasso.

**Project administration:** Nunzio Tuccitto, Giuseppe Grasso.

**Resources:** Giuseppe Grasso.

**Software:** Nunzio Tuccitto.

**Supervision:** Nunzio Tuccitto, Giuseppe Grasso.

**Validation:** Nunzio Tuccitto.

**Visualization:** Giuseppe Grasso.

**Writing – original draft:** Gabriele Antonio Zingale, Irene Pandino, Damiano Calcagno, Nunzio Tuccitto, Giuseppe Grasso.

**Writing – review & editing:** Giuseppe Grasso.

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
