## [Decision Letter · Decision Letter 0]

10 Sep 2024

PONE-D-24-24894Label-Free determination of diffusion coefficients at the nanoscale through a Surface Plasmon Resonance-Based approachPLOS ONE

Dear Dr. Grasso,

Thank you for submitting your manuscript to PLOS ONE. After careful consideration, we feel that it has merit but does not fully meet PLOS ONE’s publication criteria as it currently stands. Therefore, we invite you to submit a revised version of the manuscript that addresses the points raised during the review process.

**ACADEMIC EDITOR: **The authors refined using a commercial SPR instrument to determine the diffusion coefficient of substances, including those undergoing complex oligomerization. The manuscript has good explanations and details of all calculations + modeling. Similar approaches exist for instruments based on other principles, for example using a UV-selective detection system (e.g., Malvern Viscosizer). Thus, the strategy is only partially novel. However, the refinements for large molecules and subsequent illustrations with insulin oligomerization are quite impressive. I appreciate the authors' efforts in making the manuscript clear and precise. The authors should expand on the pros and cons of D-SPR vs. other detection methods (also based on Taylor dispersion analysis).==============================

We look forward to receiving your revised manuscript.

Kind regards,

Hemant Kumar Daima, PhD

Academic Editor

PLOS ONE

Journal Requirements: When submitting your revision, we need you to address these additional requirements. 1. Please ensure that your manuscript meets PLOS ONE's style requirements, including those for file naming. The PLOS ONE style templates can be found at https://journals.plos.org/plosone/s/file?id=wjVg/PLOSOne_formatting_sample_main_body.pdf and https://journals.plos.org/plosone/s/file?id=ba62/PLOSOne_formatting_sample_title_authors_affiliations.pdf 2. We note you have not yet provided a protocols.io PDF version of your protocol and/or a protocols.io DOI. When you submit your revision, please provide a PDF version of your protocol as generated by protocols.io (the file will have the protocols.io logo in the upper right corner of the first page) as a Supporting Information file. The filename should be S1_file.pdf, and you should enter “S1 File” into the Description field. Any additional protocols should be numbered S2, S3, and so on. Please also follow the instructions for Supporting Information captions [https://journals.plos.org/plosone/s/supporting-information#loc-captions]. The title in the caption should read: “Step-by-step protocol, also available on protocols.io.” Please assign your protocol a protocols.io DOI, if you have not already done so, and include the following line in the Materials and Methods section of your manuscript: “The protocol described in this peer-reviewed article is published on protocols.io (https://dx.doi.org/10.17504/protocols.io.[...]) and is included for printing purposes as S1 File.” You should also supply the DOI in the Protocols.io DOI field of the submission form when you submit your revision. If you have not yet uploaded your protocol to protocols.io, you are invited to use the platform’s protocol entry service [https://www.protocols.io/we-enter-protocols] for doing so, at no charge. Through this service, the team at protocols.io will enter your protocol for you and format it in a way that takes advantage of the platform’s features. When submitting your protocol to the protocol entry service please include the customer code PLOS2022 in the Note field and indicate that your protocol is associated with a PLOS ONE Lab Protocol Submission. You should also include the title and manuscript number of your PLOS ONE submission. 3. Thank you for stating the following financial disclosure: "This research was supported by MIUR, PRIN: P2022AW2H9 “Molecular details on the early phase of amyloid beta peptides aggregation: a multilevel approach based on carbon dots fluorescence and diffusion coefficients measurements to unveil the pathogenic molecular mechanisms at the base of Alzheimer’s disease” and “Progetto Pharma-HUB - HUB per il riposizionamento di farmaci nelle malattie rare del sistema nervoso in età pediatrica” (T4-AN-04). I. Pandino, G.A. Zingale and M. L. Perina were supported by the PhD program in Chemical Sciences, University of Catania. The authors acknowledge the Ministry of Health and Fondazione Roma. The authors further acknowledge LazioInnova for its financial support (grant: A0375-2020-36591)." Please state what role the funders took in the study.  If the funders had no role, please state: ""The funders had no role in study design, data collection and analysis, decision to publish, or preparation of the manuscript."" If this statement is not correct you must amend it as needed. Please include this amended Role of Funder statement in your cover letter; we will change the online submission form on your behalf. 4. Thank you for stating the following in the Acknowledgments Section of your manuscript: "This research was supported by MIUR, PRIN: P2022AW2H9 “Molecular details on the early phase of amyloid beta peptides aggregation: a multilevel approach based on carbon dots fluorescence and diffusion coefficients measurements to unveil the pathogenic molecular mechanisms at the base of Alzheimer’s disease” and “Progetto Pharma-HUB - HUB per il riposizionamento di farmaci nelle malattie rare del sistema nervoso in età pediatrica” (T4-AN-04). G.A. Zingale, I. Pandino and M. L. Perina were supported by the PhD program in Chemical Sciences, University of Catania. The authors acknowledge the Ministry of Health and Fondazione Roma. The authors further acknowledge LazioInnova for its financial support (grant: A0375-2020-36591). Figures were created with icons from BioRender.com." We note that you have provided funding information that is not currently declared in your Funding Statement. However, funding information should not appear in the Acknowledgments section or other areas of your manuscript. We will only publish funding information present in the Funding Statement section of the online submission form. Please remove any funding-related text from the manuscript and let us know how you would like to update your Funding Statement. Currently, your Funding Statement reads as follows: "This research was supported by MIUR, PRIN: P2022AW2H9 “Molecular details on the early phase of amyloid beta peptides aggregation: a multilevel approach based on carbon dots fluorescence and diffusion coefficients measurements to unveil the pathogenic molecular mechanisms at the base of Alzheimer’s disease” and “Progetto Pharma-HUB - HUB per il riposizionamento di farmaci nelle malattie rare del sistema nervoso in età pediatrica” (T4-AN-04). I. Pandino, G.A. Zingale and M. L. Perina were supported by the PhD program in Chemical Sciences, University of Catania. The authors acknowledge the Ministry of Health and Fondazione Roma. The authors further acknowledge LazioInnova for its financial support (grant: A0375-2020-36591)." Please include your amended statements within your cover letter; we will change the online submission form on your behalf. 5. Thank you for stating the following in your Competing Interests section: "None" Please complete your Competing Interests on the online submission form to state any Competing Interests. If you have no competing interests, please state ""The authors have declared that no competing interests exist."", as detailed online in our guide for authors at http://journals.plos.org/plosone/s/submit-now   This information should be included in your cover letter; we will change the online submission form on your behalf. 6. Please include a copy of Table 5 which you refer to in your text on page 16 in PDF submission.

Reviewers' comments:

Reviewer's Responses to Questions

**Comments to the Author**

1. Does the manuscript report a protocol which is of utility to the research community and adds value to the published literature?

Reviewer #1: Yes

Reviewer #2: Yes

Reviewer #3: Yes

2. Has the protocol been described in sufficient detail?

To answer this question, please click the link to protocols.io in the Materials and Methods section of the manuscript (if a link has been provided) or consult the step-by-step protocol in the Supporting Information files.

The step-by-step protocol should contain sufficient detail for another researcher to be able to reproduce all experiments and analyses.

Reviewer #1: Yes

Reviewer #2: No

Reviewer #3: Yes

3. Does the protocol describe a validated method?

Reviewer #1: Yes

Reviewer #2: No

Reviewer #3: Yes

4. If the manuscript contains new data, have the authors made this data fully available?

Reviewer #1: Yes

Reviewer #2: N/A

Reviewer #3: N/A

**5. Is the article presented in an intelligible fashion and written in standard English?**

Reviewer #1: Yes

Reviewer #2: Yes

Reviewer #3: Yes

6. Review Comments to the Author

Reviewer #1: Dear Authors,

The protocol "Label free determination of diffusion co-efficient at the nanoscale through surface plasmon resonance based approach" defines a method for enhancing the limitations of a radial flow observed by most of the fluids. The normal Taylor's diffusion measurements for diffusion co-efficients through UV-vis spectroscopic methods has the limitations of analyzing a complex fluids and in case of highly concentrated fluids, the intramolecular interactions are neglected. Therefore, the authors have designed a protocol using python programming to include the skewness of the data and have obtained improved diffusion co-efficients of various different fluids, including biomolecules. The manuscript is well written

These are some of my observations

1. The title can be made more appropriate including the unique feature of the protocol- python programming

2. The keywords are redundant for eg: proteins, amino acids- This section can include more appropriate keywords and reduce the number of keywords

3. Primarily, the whole protocol is an improvement of Taylor's law of diffusion. Therefore, there needs to be detailed explanation about the law

4. The importance of including the corrections in diffusion readings needs to be elaborated.

Reviewer #2: Here are some suggestions for improving this work's quality:

1. Discuss potential concerns with analyte adherence to the SPR gold chip and suggest minimising techniques.

2. Investigate further applications for the approach beyond those described, notably in drug discovery and protein engineering.

3. Compare the suggested method to other diffusion coefficient measurement techniques to highlight its advantages and limitations.

4. Determine the sensitivity of the SPR-based approach in detecting minor changes in diffusion coefficients and how this affects its broader applicability.

5. Include a discussion of the newly developed WaveFlex Biosensors, comparing their capabilities to the suggested SPR-based technique.

Reviewer #3: Authors should check for some minor grammatical error in the abstract in line 4 and line 10 as well as in the subsequent sections of the main manuscript.

7. PLOS authors have the option to publish the peer review history of their article (what does this mean?). If published, this will include your full peer review and any attached files.

Reviewer #1: No

Reviewer #2: No

Reviewer #3: No

---

## [Author Response · Author response to Decision Letter 0]

1 Oct 2024

ACADEMIC EDITOR: 

The authors refined using a commercial SPR instrument to determine the diffusion coefficient of substances, including those undergoing complex oligomerization. The manuscript has good explanations and details of all calculations + modeling. Similar approaches exist for instruments based on other principles, for example using a UV-selective detection system (e.g., Malvern Viscosizer). Thus, the strategy is only partially novel. However, the refinements for large molecules and subsequent illustrations with insulin oligomerization are quite impressive. I appreciate the authors' efforts in making the manuscript clear and precise. 

The authors should expand on the pros and cons of D-SPR vs. other detection methods (also based on Taylor dispersion analysis).

Dear Academic Editor, thank you for your feedback and appreciation of our efforts to enhance the clarity and precision of our manuscript. We have carefully considered your suggestion to expand on the pros and cons of D-SPR in comparison to other detection methods, including those based on Taylor dispersion analysis. As you noted, this was also a recommendation from Reviewer #2. In response to these comments, we have expanded our discussion on the advantages and limitations of D-SPR compared to alternative techniques, focusing on the unique capabilities of D-SPR in terms of its label-free nature. We believe that this revised section offers a comprehensive overview of the pros and cons of D-SPR, enabling readers to better understand its potential and limitations within the context of our research.

Reviewer #1: 

Dear Authors,

The protocol "Label free determination of diffusion co-efficient at the nanoscale through surface plasmon resonance based approach" defines a method for enhancing the limitations of a radial flow observed by most of the fluids. The normal Taylor's diffusion measurements for diffusion co-efficients through UV-vis spectroscopic methods has the limitations of analyzing a complex fluids and in case of highly concentrated fluids, the intramolecular interactions are neglected. Therefore, the authors have designed a protocol using python programming to include the skewness of the data and have obtained improved diffusion co-efficients of various different fluids, including biomolecules. The manuscript is well written

These are some of my observations

1. The title can be made more appropriate including the unique feature of the protocol- python programming

We have corrected the title according to the Reviewer’s indication.

2. The keywords are redundant for eg: proteins, amino acids- This section can include more appropriate keywords and reduce the number of keywords

We have corrected the keywords according to the Reviewer’s indication.

3. Primarily, the whole protocol is an improvement of Taylor's law of diffusion. Therefore, there needs to be detailed explanation about the law

This is an article exclusively about the model, so it is not our aim to provide too much detail on the theoretical background of the protocol. However, we agree with the reviewer that some clarifying information improves understanding. Accordingly, we have included the following paragraph in the revised version of the article. We believe that this revised explanation addresses the concerns raised and provides a more thorough grounding of the theory behind the protocol.

‘The measurement of the diffusion coefficient D in the protocol presented here is based on Taylor's law of diffusion. This method is widely used to estimate the D value of molecules or particles travelling in confined spaces, such as capillary tubes. The principle was first introduced by Sir Geoffrey Taylor in 1953 and later extended to describe the dispersion of solutes in a laminar flow through cylindrical tubes. Taylor-Aris dispersion, as it is often called, combines the effects of molecular diffusion to analyse the longitudinal diffusion of solute plugs in a tube. The basis of Taylor's law is the observation that solutes moving in a fluid medium undergo both axial (along the length of the tube) and radial (across the cross-section of the tube) diffusion. In laminar flow, molecules near the centre of the tube move faster than those near the walls, due to the parabolic velocity profile typical of fluid flow in capillaries. This difference in flow velocity causes solutes to disperse longitudinally. However, radial diffusion (which is perpendicular to the direction of flow) acts to mix the faster-moving molecules in the central regions with slower-moving molecules near the tube walls, moderating longitudinal dispersion over time. Taylor's analysis showed that when molecules undergo random movement (radial diffusion), this leads to a concentration profile along the length of the tube, which eventually widens due to axial dispersion. Taylor derived an effective longitudinal diffusion coefficient that incorporates both molecular diffusion and convective flow effects. In our experimental protocol, Taylor's diffusion analysis is used to estimate the diffusion coefficient of solute molecules by analysing their diffusion behaviour in the capillary tube. In this case, a microfluidic system is used in which the fluid is pushed towards the microfluidic cell of the SPR detector and the interaction of the molecules with the surface plasmon resonance interface (SPR) is detected. Taylor's law provides a framework for understanding the coupling between convective drift and molecular diffusion in a capillary tube. This interaction is fundamental to analysing solute diffusion, allowing us to calculate an effective diffusion coefficient that takes into account both fluid velocity and the molecular properties of the solute. The assumption that radial diffusion dominates over axial diffusion allows us to simplify the analysis and treat the system with one-dimensional approximations. This greatly reduces the complexity of the model while still providing accurate results for the diffusion coefficient.’

4. The importance of including the corrections in diffusion readings needs to be elaborated.

We agree with the reviewer that some clarifying information improves understanding. Accordingly, we have included the following paragraph in the revised version of the article.

“Taylor diffusion analysis offers a reliable framework for measuring the diffusion coefficients D of solute molecules in capillary tubes. However, the method assumes idealised conditions, such as a fully developed flow profile and an adequate capillary length to ensure the correct behaviour of solutes in both radial and axial diffusion. In practice, the finite tube length and high flow velocity often lead to deviations from ideal assumptions, which introduce asymmetries in the detected concentration profile. Recognising and correcting these deviations is crucial to ensure the accuracy of the derived D values. Under the experimental conditions under which the protocol is performed, one of the main causes of skewness or asymmetry in the diffusion curve is related to insufficient tube length and high flow velocity. When the capillary tube is too short in relation to the velocity of the carrier fluid, the solute cap does not have sufficient time to diffuse completely radially before reaching the detector. This incomplete radial mixing results in an elongated and asymmetric concentration profile, instead of the ideal Gaussian shape expected under full Tayloristic diffusion conditions. The short tube length limits the time available for solutes to equilibrate radially, causing a distortion of the concentration profile detected by the SPR sensor. The relatively high flow velocity exacerbates the problem by rapidly conveying solutes along the axial direction, further reducing the effective radial diffusion time. Both factors result in a non-Gaussian profile at the detector, with an obvious tailing effect on the signal. Given these non-idealities, it is essential to apply appropriate correction factors to ensure that the derived scattering coefficients are accurate. The corrections help to account for signal skewness due to imperfect mixing of solutes within the capillary. When the concentration profile at the detector is skewed due to short tube length and high flow velocity, a correction is applied to the measured variance of the derivative before the concentration profile. This adjustment compensates for the fact that the solute molecules have not undergone complete radial diffusion, which would normally result in a symmetrical, Gaussian profile. The correction factor adjusts the variance according to the ratio of tube length to flow velocity, ensuring that the diffusion coefficient.’

Reviewer #2: 

Here are some suggestions for improving this work's quality:

1. Discuss potential concerns with analyte adherence to the SPR gold chip and suggest minimising techniques.

We thank the Reviewer for this valuable comment regarding analyte interaction with the gold surface of the sensor-chip. As correctly noted, this is a potential concern that we addressed in detail in Chapter 4 “Functionalization of the gold sensor-chip” of the supplementary information (S_1 file). However, we have included a concise comment on this topic in the method validation section of the main manuscript, directing readers to the Supplementary Information for a more comprehensive discussion. We believe that the strategies outlined, including the evaluation of the molecular surface charge under the chosen experimental conditions, effectively mitigate this issue and ensure reliable experimental results. 

2. Investigate further applications for the approach beyond those described, notably in drug discovery and protein engineering.

We thank the Reviewer for the relevant feedback. We have taken the suggestion into consideration and have expanded upon the potential applications of our approach in the conclusion section of the manuscript. We believe that the additional discussion on drug discovery and protein engineering provides a more comprehensive overview of the broader implications of our work.

3. Compare the suggested method to other diffusion coefficient measurement techniques to highlight its advantages and limitations.

We appreciate the Reviewer’s suggestion to compare our proposed method to other diffusion coefficient measurement techniques. We have already conducted a thorough comparison in the introduction section of our manuscript. To further emphasize the unique advantages of our approach, we highlighted its label-free nature, which eliminates the need for artificial modifications that could potentially alter the molecular structure and properties. We also highlighted its fast measurement times and simple data analysis using a Python script, making it highly efficient and user-friendly compared to more traditional techniques.

4. Determine the sensitivity of the SPR-based approach in detecting minor changes in diffusion coefficients and how this affects its broader applicability.

We appreciate the Reviewer comment regarding the sensitivity of our SPR-based approach. We have expanded the conclusions section to provide a more detailed analysis of the method’s ability to detect minor changes in diffusion coefficients and its potential applications.

5. Include a discussion of the newly developed WaveFlex Biosensors, comparing their capabilities to the suggested SPR-based technique.

We have added a small paragraph in the introduction to mention the WaveFlex Biosensors. However, a full comparison between our experimental approach and the newly developed WaveFlex Biosensors is not feasible because our method cannot be used for the detection of an analyte based on a biomolecular interaction but it is rather useful for the estimation of the diffusion coefficient of a single molecule not interacting with the SPR surface.

Reviewer #3: 

Authors should check for some minor grammatical error in the abstract in line 4 and line 10 as well as in the subsequent sections of the main manuscript.

We have corrected the text according to the Reviewer’s indication.

---

## [Editor Report · Decision Letter 1]

10 Oct 2024

Label-Free determination of diffusion coefficients at the nanoscale through modelling of the Surface Plasmon Resonance signal

PONE-D-24-24894R1

Dear Dr. Grasso,

We’re pleased to inform you that your manuscript has been judged scientifically suitable for publication and will be formally accepted for publication once it meets all outstanding technical requirements.

Kind regards,

Hemant Kumar Daima

Academic Editor

PLOS ONE
---

## [Editor Report · Acceptance letter]

5 Nov 2024

PONE-D-24-24894R1 

PLOS ONE

Dear Dr. Grasso, 

I'm pleased to inform you that your manuscript has been deemed suitable for publication in PLOS ONE. Congratulations! Your manuscript is now being handed over to our production team.

Kind regards, 

on behalf of

Dr. Hemant Kumar Daima 

Academic Editor

PLOS ONE